# Copresentation of BMP-6 and RGD Ligands Enhances Cell Adhesion and BMP-Mediated Signaling

**DOI:** 10.3390/cells8121646

**Published:** 2019-12-15

**Authors:** Francesca Posa, Anna Luise Grab, Volker Martin, Dirk Hose, Anja Seckinger, Giorgio Mori, Slobodan Vukicevic, Elisabetta Ada Cavalcanti-Adam

**Affiliations:** 1Department of Cellular Biophysics, Max Planck Institute for Medical Research, Jahnstr. 29, 69120 Heidelberg, Germany; 2Department of Biophysical Chemistry, Institute of Physical Chemistry, Heidelberg University, Im Neuenheimer Feld 253, 69120 Heidelberg, Germany; 3Department of Clinical and Experimental Medicine, University of Foggia, via L. Pinto, 71122 Foggia, Italy; 4Genome Biology Unit, EMBL, Meyerhofstraße 1, 69117 Heidelberg, Germany; 5Laboratory for Myeloma Research and Medical Clinic V, University Hospital Heidelberg, Im Neuenheimer Feld 410, 69120 Heidelberg, Germany; 6Laboratory for Mineralized Tissues, Center for Translational and Clinical Research, School of Medicine, University of Zagreb, Šalata 11, 10000 Zagreb, Croatia

**Keywords:** integrin ligands, bone morphogenetic protein 6, surface copresentation, myotube formation, BMP/Smad signaling

## Abstract

We report on the covalent immobilization of bone morphogenetic protein 6 (BMP-6) and its co-presentation with integrin ligands on a nanopatterned platform to study cell adhesion and signaling responses which regulate the transdifferentiation of myoblasts into osteogenic cells. To immobilize BMP-6, the heterobifunctional linker MU-NHS is coupled to amine residues of the growth factor; this prevents its internalization while ensuring that its biological activity is maintained. Additionally, to allow cells to adhere to such platform and study signaling events arising from the contact to the surface, we used click-chemistry to immobilize cyclic-RGD carrying an azido group reacting with PEG-alkyne spacers via copper-catalyzed 1,3-dipolar cycloaddition. We show that the copresentation of BMP-6 and RGD favors focal adhesion formation and promotes Smad 1/5/8 phosphorylation. When presented in low amounts, BMP-6 added to culture media of cells adhering to the RGD ligands is less effective than BMP-6 immobilized on the surfaces in inducing Smad complex activation and in inhibiting myotube formation. Our results suggest that a local control of ligand density and cell signaling is crucial for modulating cell response.

## 1. Introduction

Bone morphogenetic proteins (BMPs) are growth factors belonging to the TGF-β superfamily. They exert pleiotropic effects and BMP-2 and -7 have been approved in clinical use for their potent induction of bone formation [1]. Among the BMPs, BMP-6 is able to mediate osteogenic differentiation and is currently tested in clinical trials in combination with an autologous blood coagulum [2,3] showing efficacy in in vivo studies at low amounts, e.g., 50 μg in whole blood containing devices. These doses are much lower when compared to the ones (e.g., 3.5 mg in a bovine collagen carrier) used for BMP-2 and -7 [4,5]. Thus, BMP-6 is an attractive candidate for future development in bone tissue engineering.

First validated in C2C12 mouse myoblasts [6], and then further investigated in a variety of primary human cells [7,8], the binding profile of BMP-6 to the type I (ALK-2 strongly and weakly to ALK-6) and type II receptors, mirrors, to some extent, the one of BMP-7. While BMP-7 is emerging as a regulator of cell adhesion and migration by modulating β-integrin activation [9] and by inducing the expression of α5 integrins [10], the interplay between BMP-6-mediated signaling and cell adhesion to the extracellular matrix remains largely unexplored. When combined with fibronectin, BMP-6 osteogenic activity is prolonged [11], suggesting that BMP-6 matrix interactions participate in the regulation of cell fate. Furthermore, BMP-6 signaling activity, which results in the phosphorylation of the Smad 1/5/8 complex, appears to be modulated by the interactions of the growth factor with heparin. When bound to heparin, BMP-6 is able to induce osteogenic responses both in vitro and in vivo [12]. Prediction of heparin binding sites in BMPs have identified the N-terminus region as highly reactive [13,14]. Additionally, the binding of BMP-6 and the nature of its interactions within the extracellular matrix make the binding of its endogenous inhibitors less effective and this in turn might prolong its signaling [6,15,16,17].

Here, we developed a setup to mimic BMP-6 matrix-bound form based on the copresentation of BMP-6 and integrin ligands (RGD, arginine-glycine-aspartic acid). The setup is based on surface nanopatterning of gold particles by diblock copolymer micellar nanolithography surrounded by a protein repellent layer of polyethylene glycol (PEG) [18] and functionalization with RGD ligands and BMP-6. When bound to gold nanoparticles, the adhesive ligand is presented in ordered hexagonal patterns at spacing ranging from 50 to 100 nm [19]. This is known to regulate integrin clustering and favor cell adhesion for spacing ≤58 nm [20]. Alternatively, RGD is immobilized on a PEG layer at a surface density of 1 mol% as reported in Schenk et al. [21], corresponding to a surface density of 0.53 ng/cm^2^. This results in disordered presentation, which may lead to a shift in spacing threshold for cell adhesion [22]. Our aim is to determine the effects of BMP-6 and integrin-mediated adhesion on BMP-mediated signaling and cell differentiation. We previously showed that BMP-2 immobilized via its amine residues is still bioactive and that it signals even in nanograms to cells, due to its sustained presentation when compared to BMP-2 added to the culture media [23,24]. In the present work, we apply the same immobilization strategy to BMP-6 homodimers and investigate signaling in C2C12 cells. Single BMP-6 molecules are selectively immobilized on nanoparticles to control growth factor surface density at the nanoscale. With our setup, which is based on dual functionalization chemistry for immobilizing two different types of ligands, we achieved here spatial control over the copresentation of growth factors and integrin ligands to study cell signaling and adhesion for BMP-6 at in low amounts. We present a promising approach for biomaterials functionalization aiming at controlling cell fate through adhesion at the material interface.

## 2. Materials and Methods

### 2.1. Preparation of Homogeneous Coated and Nanopatterned Surfaces

In order to produce gold homogeneous coated surfaces, clean glass coverslips (20 × 20 mm) were coated with a 10 nm chromium layer and then a 50 nm gold layer by using a MED-020 Sputter coater (BalTec, Schalksmühle, Germany). For the nanopatterned surfaces preparation, glass coverslips were decorated with gold nanoparticles by using block copolymer micellar nanolithography (BCMN) as previously reported [19,20,24]. Substrates having an average interparticle distance of 32 ± 5 nm, 61 ± 9 nm and 110 ± 16 nm were used. For the 30 nm distance, gold-loaded polymer micelle solution of 5 mg/mL polystyrene(30,000)-b-poly-2-vinylpyridine(12,500) (Polymer Source, Quebec, Canada) in toluene with a tetrachloroauric acid to vinylpyridine monomer ratio of 0.5 was used. For 60 nm and 100 nm distances, polystyrene(110,000)-b-poly-2-vinylpyridine(52,000) and polystyrene(190,000)-b-poly-2-vinylpyridine(55,000) (Polymer Source) were used with the same concentration and monomer ratio as for the 30 nm spacing.

### 2.2. Surface Functionalization with Integrin Ligands

In the first part of our work, in order to investigate cell spreading and FAs assembly in C2C12 cells, gold nanopatterned surfaces with a distance of 50 nm were prepared as described above. The area between the gold particles was passivated with PEG2000 and cyclic-RGD peptides, c(RGDfK)-thiol, were immobilized to the gold particles. Each nanoparticle serves as binding site for only one integrin receptor, because of its limited size (experimental setup shown in Figure 1a).

To introduce adhesive ligands in the passivated background of the nanopapatterned surfaces (as shown in Figure 4a), PEG2000 silane was mixed with PEG3000 silane carrying alkyne groups at 99:1 ratio. The cyclic peptide c(RGDfE)K(N_3_) (PSL Peptide Specialty Laboratories GmbH, Heidelberg, Germany) was bound to the PEG layer via copper(I)-catalyzed-azide-alkyne cycloaddition as described in [21].

### 2.3. Surface Immobilization of BMP-6

To immobilize BMP-6 on the substrates, gold-coated and nanopatterned glass substrates were then incubated with a 1 mM solution of the hetero-bifunctional linker 11-mercaptoundecanoyl-*N*-hydroxysuccinimide ester (MU-NHS, ProChimia Surfaces, Gdynia, Poland) in DMF for 4 h at room temperature. After sonication, the substrates were rinsed with DMF and MeOH.

Lyophilized carrier-free, glycosylated GMP-produced rhBMP-6 expressed in mammalian cells (Genera Research, Rakov Potok, Croatia) was reconstituted in sterile 4 mM HCl to obtain a 100 μg/mL stock solution. This solution was diluted in phosphate-buffered saline (PBS) (Gibco BRL) containing 1 M NaCl (Carl Roth) and adjusted to pH 8.5 immediately prior to use, in order to obtain a working concentration of 3.5 μg/mL. The NHS-functionalized substrates were incubated with the BMP-6 solution at 4 °C for 14 h. The surfaces were sonicated in PBS (1 M NaCl) and washed three times with the same buffer to remove unbound proteins. The surface concentration of BMP-6 on the homogeneous surfaces was approximately of 70–80 ng/cm^2^ as we previously showed for BMP-2 [23].

Concerning the nanopatterned surfaces, the amount of immobilized protein was estimated by quantifying the average number of gold nanoparticles per μm^2^ from scanning electron microscopy (SEM) images of the samples. Based on previous studies on the quantification of growth factor binding to nanopatterned substrates [24], a coverage of approx. 90% of the particles with a single BMP-6 homodimer leads to surface density of immobilized proteins of 4.8 ng/cm^2^ (corresponding to an amount of 19 ng per sample), 1.6 ng/cm^2^ (6.3 ng per sample) and 0.4 ng/cm^2^ (1.4 ng per sample), respectively, for the different interparticle spacing (30, 50 and 100 nm).

### 2.4. Analysis of Surface Binding of BMP-6

Quarz crystal microbalance with dissipation monitoring (QCM-D) measurements were performed on a Q-Sense E4 system (Q-Sense) with gold crystals (Q-sense) in an open module (Q-sense). The crystals were functionalized with the MU-NHS linker as mentioned previously. All measurements were performed in a volume of 200 μL and at room temperature. For the QCM-D experiments, the rinsing consisted of buffer 1 as baseline (PBS, pH 7.4) and buffer 2 (PBS, pH 8.5, 1 M NaCl; cPBS). Then, 1 μg/mL rhBMP-6 was incubated for 60 min in buffer 2 followed by rinsing in buffer 2 and 1 for 10 min each. Bovine serum albumin (BSA) blocking was performed for 60 min, followed by a 30-min incubation with anti-rhBMP-6 mouse IgG (5 μg/mL), rinsing and incubation with anti-mouse IgG (5 μg/mL).

### 2.5. Cell Culture

Mouse myoblasts C2C12 (ATCC CRL-1772), when grown in vitro, spontaneously fuse and form myotubes; they transdifferentiate into cells of osteoblastic lineage in responses to BMPs [25]. C2C12 cells were cultured as subconfluent monolayers in growth medium, consisting of Dulbecco’s modified Eagle’s medium (DMEM) (Thermo Fisher Scientific, Waltham, MA, USA) supplemented with 10% heat inactivated fetal bovine serum (FBS) (Biochrom GmbH, Berlin, Germany) and 1% penicillin/streptomycin (Thermo Fisher Scientific) at 37 °C and 5% CO_2_.

### 2.6. Western Blot Analysis of Protein Expression

Western blot analyses were performed as described in [24]. Briefly, cells were starved in serum free DMEM for at least 5 h and then seeded at a density of 80 × 10^3^ cells/cm^2^ in DMEM without FBS. After the different time points, cells were lysed in RIPA lysis buffer (Sigma Aldrich, St. Louis, MO, USA) containing protease inhibitors (Roche, Basel, Switzerland). Cell lysates were separated by SDS-PAGE, transferred to nitrocellulose membrane (Thermo Fisher Scientific) and then probed with 1:1000 rabbit anti-pSMAD 1/5/8 (Cell signaling technology, Danvers, MA, USA) and 1:2000 mouse anti-β-actin (Sigma Aldrich) antibodies. After incubation with 1:5000 goat anti-rabbit or anti-mouse HRP-conjugated IgG (Santa Cruz Biotechnology, Dallas, TX, USA) the detection was performed with an ECL Plus Detection Kit and an Amersham Imager 600 (GE Healthcare, Chicago, IL, USA).

### 2.7. Immunostaining and Microscopy

After starving for 5 h, C2C12 cells were trypsinized and 10 × 10^3^ cells/cm^2^ were seeded on the different substrates. Control cells were not treated with the growth factor, treated cells were exposed to surface immobilized BMP-6 (iBMP-6) or to BMP-6 added to the culture media (sBMP-6) under low serum conditions (2% FBS). For focal adhesion staining, cells were fixed after 4 h with 4% (*w*/*v*) paraformaldehyde (PFA) in PBS. Cells were permeabilized with 0.1% (*v*/*v*) Triton-X-100 in PBS for 5 min, washed and incubated with 1% BSA for 1 h to avoid non-specific protein binding. The following antibodies were used: vinculin (Sigma Aldrich) 1:100, followed by anti-mouse-FITC secondary antibody (1:100) and Phalloidin-TRITC (0.2 μg/mL) at RT in 1% BSA for 1 h. Samples were embedded in Mowiol containing 0.1% (*v*/*v*) DAPI for an additional staining of the nucleus.

The formation of myotubes was analyzed by staining on day 4. Cells were cultured with DMEM supplemented with 2% FBS and the media was changed after three days without adding any additional sBMP-6. After 4 days, cells were fixed and permeabilized for immunofluorescence staining with monoclonal anti-myosin heavy chain (MHC, MF20, Developmental Studies Hybridoma Bank, University of Iowa; 5 μg/mL). Cells were labeled with Alexa 488 anti-mouse IgG antibody (Invitrogen, Carlsbad, CA, USA) and DAPI solution (Sigma).

Samples were imaged by immunofluorescence microscopy (DeltaVision (DV) microscope (Applied Precision Inc., Woodbridge, Canada; data processing controlled by Resolve 3D (Applied Precision Inc., Issaquah, WA, USA)) equipped with a CCD camera (Olympus, Shinjuku, Tokyo, Japan) and using a 60× oil objective for imaging of cell cytoskeleton and focal adhesions, and a 10× air objective for imaging of myotubes. The images were adjusted in brightness and color with ImageJ software (Research Services Branch, Image Analysis Software Version 1.52c, NIH, Bethesda, MD, USA).

### 2.8. Statistical Analysis

Statistical analysis was performed using the GraphPad Prism software (San Diego, CA, USA). Groups were compared using two-sample *t*-test with *p*-values < 0.05 considered as statistically significant (indicated as # *p* < 0.05, § *p* < 0.01, * *p* < 0.001). ImageJ was used to process images. Western blot bands and myotubes area were quantified using ImageJ [26]. All plotted data show mean values with standard deviations calculated from at least three independent experiments (samples in duplicates or triplicates).

## 3. Results

### 3.1. BMP-6 Influences Cell Spreading and Focal Adhesion Assembly

To evaluate the influence of BMP-6 on cell adhesion and spreading, C2C12 cells were cultured for 4 h on nanopatterned surfaces presenting RGD ligands at 50 nm spacing in absence of BMP-6 (No BMP-6) or in presence of the growth factor added to the media (sBMP-6) (Figure 1a). The 50 nm spacing has been reported to promote cell spreading and focal adhesion assembly [20]. As expected, RGD-ligands allowed integrin mediated cell adhesion and spreading of C2C12 cells (Figure 1b). Comparing the formation of focal adhesions (FAs) and total cell area of samples treated or not with BMP-6 we found that FAs were larger and increased in presence of BMP-6. Interestingly, cells in presence of both RGD and sBMP-6 showed two-fold reduction in cell spreading (Figure 1c). This suggests that BMP-6 acts on cell cytoskeletal tension, which is tightly couple with BMP-induced osteogenic signaling [27].

### 3.2. Surface Immobilized BMP-6 is Not Internalized by Cells and Triggers Smad Signaling

To study the effect of binding proximity of integrins and BMP receptors on C2C12 adhesion and signaling which regulate cell fate, we applied a selective chemistry approach, using a self-assembled monolayer of the heterobifunctional linker mercaptoundecanoic-*N*-hydroxysuccinimide ester (MU-NHS) to immobilize BMP-6 on gold surfaces at controlled surface density of 80 ng/cm^2^ as previously shown with BMP-2 [23,24,28]. We used quartz crystal microbalance with dissipation monitoring (QCM-D) to demonstrate the validity of our platform by monitoring the binding kinetics of BMP-6 to the MU-NHS linker and its binding specificity (Figure A1(1,2)).

To visualize the growth factor on surfaces, we detected the surface immobilized BMP-6 (iBMP-6) by fluorescence microscopy (Figure 2a). We used four different glass coverslips (Figure 2a); in brief, a (i) glass coated with a 50-nm homogeneous gold layer by metal sputtering (indicated as Au), (ii) glass uncoated (indicated as Glass), (iii) glass coated with PEG (indicated as PEG), (iv) glass decorated with 8-nm size gold nanoparticles having a spacing of 30 nm arranged in hexagonal lattice, with the interparticle surface covered by PEG to prevent unspecific binding of protein (indicated as PEG/AuNP) (Figure 2a). The sample was first incubated with the MU-NHS linker and then with BMP-6, followed by detection with antibodies. Indirect immunofluorescence staining shows signal on Au, glass and PEG/AuNP whereas on PEG coated surfaces no binding takes place, proving that PEG completely prevents the unspecific deposition of the growth factor and antibodies on the surface, which otherwise takes place on glass. Additionally, on PEG/AuNP, the staining for BMP-6 is rather homogeneous in comparison to Au coated, further indicating the binding specificity of BMP-6 and the antibodies. Furthermore, the bond between the NHS group of the linker and the amine residues of the growth factor is stable, even after washing and further treatment with detection reagents.

We next determined the stability of the chemical bond of BMP-6 to the surface in presence of cells (Figure 2b). C2C12 were cultured in tissue culture dishes and approached from the top [23] with (i) glass coverslips without growth factor (control, No BMP-6), (ii) glass coverslips functionalized with iBMP-6, (iii) glass coverslips in presence of BMP-6 added to the culture media (sBMP-6). The surfaces were in contact to cells for different time points (30, 60 and 120 min) and detection of remaining BMP-6 on the samples was done by chemiluminescence (Figure 2b). BMP-6 was labeled with primary anti-BMP-6 antibody and secondary antibody conjugated with HRP, then detected by chemiluminescence. In samples where the growth factor is absent or added to the culture media, BMP-6 is not detected neither before nor after incubation with cells. When BMP-6 is covalently immobilized on the surface (iBMP-6), it is detected after presentation to cells, indicating that cells could not remove it.

To determine if iBMP-6 retains its biological activity, we next performed Western blotting to detect Smad1/5/8 phosphorylation levels (Figure 2c). The activation of receptor-regulated Smads by BMP-6 converts the myogenic differentiation of C2C12 cells into osteogenic [6]. Exposing cells to control surfaces without BMP-6 does not lead to BMP-mediated signaling. In cells exposed for 30, 60 or 120 min to 80 ng/cm^2^ of BMP-6 (calculated according to molecular size and weight to obtain complete surface coverage [23]) either added to the media (sBMP-6) or immobilized on the surface (iBMP-6), SMAD 1/5/8 is phosphorylated and no statistically significant differences can be observed between the two groups. The peak of short term signaling, i.e., Smad 1/5/8 phosphorylation, has been reported to be at 120 min when the growth factor is immobilized on the surfaces and at 60 min after addition of sBMP-6 to cell cultures.

### 3.3. Surface Copresentation of Immobilized BMP-6 and RGD Ligands Promotes Adhesion and Induces BMP-Mediated Cell Responses

Although our system based on gold coating and self-assembled monolayer of MU-NHS linkers allows long term studies [23], cell adhesion to the surfaces is rather unspecific and guided by deposition of cell own matrix, which in turn mediates further signaling modulation of Smad pathways, as also reported for other systems [29,30,31]. Thus, to create an adhesive background, while preserving the setup for the presentation of iBMP-6 covalently bound to the surface, we further developed the nanopatterned surfaces (PEG/AuNP) shown in Figure 3a by functionalizing the PEG layer with c(RGDfE)K(N_3_), an integrin binding peptide. This setup is adapted from Schenk et al., where the dual functionalization of cRGD on the gold nanoparticles and synergy peptide PHSRN on the PEG layer was developed to promote cell adhesion [21]. Following the coating with a PEG layer consisting in a mixture of PEG2000 and PEG3000-alkyne at a 99:1 ratio, the adhesive ligand carrying an azido functional group is introduced by copper(I)-catalyzed- azide-alkyne cycloaddition. Cells adhering to this surface were used as control (No BMP-6). Then, the gold nanoparticles (AuNPs) were functionalized with the MU-NHS linker so that the covalent immobilization of single BMP-6 homodimers (iBMP-6) or BMP-6 can be added to the media in soluble form (sBMP-6) (Figure 3a). The amount of cRGD immobilized on the surface is tuned by varying the number of functional groups in PEG [21] and the amount of BMPs on the surface by varying the AuNPs spacing [24], to achieve a controlled spatial distribution of the peptide and the protein, and monitor the specific response of cells to the co-presentation of biomolecule in proximity to each other.

In absence of the adhesive ligand, cells do not adhere and spread on the surfaces containing gold nanodots with an interparticle distance of 30, 50 or 100 nm, further demonstrating the passivating properties of the PEG layer. The functionalization with RGD peptides on the PEG3000-alkyne end groups promotes adhesion, as indicated by the high number of spread cells on the surfaces 3 h after seeding (Figure 3b). Regardless of the presence of BMP-6 (either added to the media or immobilized on the surface) and the amount of immobilized BMP-6, cell adhesion and spreading was comparable in presence of integrin ligands. This indicates that the immobilization of RGD on the PEG layer is fundamental for early adhesion and suggests that BMP-6-mediated cell surface interactions are not required for this process.

To study the impact of surface co-presentation of RGD and BMP-6 on cell signaling, we seeded C2C12 cells on nanopatterned surfaces and, following lysis to extract their protein content, we monitored Smad 1/5/8 phosphorylation by Western blotting 3 h after seeding (Figure 4).

C2C12 cells not treated with BMP-6 were used as control (No BMP-6), treated cells were exposed to sBMP-6 (19, 6 or 1 ng) or to the corresponding amount of the immobilized growth factor (iBMP-6, at 30, 50 or 100 nm interparticle spacing). Adhesion of cells to RGD functionalized surfaces in absence of BMP-6 does not result in Smad 1/5/8 phosphorylation, confirming the independence of Smad activation from integrin mediated signaling. When BMP-6 is either added to the media or immobilized on the surface, phosphorylation of Smad1/5/8 is observed. The phosphorylation kinetics and levels in cells adhering to RGD surfaces and exposed to different concentrations of sBMP-6 indicate that signaling takes place, but the amount of growth factor presented in the soluble form is not sufficient to trigger a strong response on 50 and 100 nm distance surfaces. Combining RGD with iBMP-6 on the surfaces enhances Smad phosphorylation levels with all the different nanoparticles surfaces and the highest pSMAD 1/5/8 expression is observable with the 30 nm surfaces. It should be noted that with our platforms we deliver here to cells 1 ng of the growth factor (when the 100 nm distance is used), which is extremely lower than the amount typically used in standard in vitro assays (20 nM, corresponding to 520 ng in 1 mL solution).

We next investigated long-term responses of cells to iBMP-6 by imaging and quantifying myotube formation in C2C12 cell cultures on the different surfaces detecting myosin heavy chain (MHC) (Figure 5a,b). In cells cultured for 6 days on Au homogeneous surfaces (Figure 5a), staining for myosin heavy chain reveals the presence of myotubes (Figure 5a, No BMP-6). When BMP-6 is either covalently immobilized on the Au surface (iBMP-6) or added to the culture media (sBMP-6), it suppresses myotube formation. Thus, iBMP-6 maintains its short- and long-term biological activity. The effective immobilization of BMP-6 on material surfaces, showing that the growth factor is still active even after a longer period of time, has not been reported thus far.

To determine the long-term effects of RGD-iBMP-6 surfaces, we cultured C2C12 cells for 4 days and performed immunofluorescence microscopy studies of myotube formation by detecting myosin heavy chain (MHC) (Figure 5b). In the control group, cells were left to adhere and further cultured on nanopatterned surfaces with clicked RGD, but in absence of BMP-6 (Figure 5b left). For the iBMP-6 and sBMP-6 groups, cells were additionally exposed to 19, 6 or 1 ng of the growth factor, either covalently immobilized on the surface (Figure 5b middle) or added to the culture media (Figure 5b right). C2C12 cells form myotubes in the control group as evidenced by the green staining for MHC; in presence of iBMP-6 and sBMP-6, MHC positive myotubes formation is significantly prevented when the factor is added at an amount of 19 ng. This inhibitory effect on myotubes formation decreases with the reduction of BMP-6 concentration used (Figure 5b). In particular, for 1 ng of sBMP-6, corresponding to the amount presented at 100 nm spacing, iBMP-6 is more effective than sBMP-6, suggesting that the sustained presentation of the growth factor, activates signaling pathways even after several days in culture. As such, surface copresentation of low amounts of BMP-6 together with integrin ligands guides cell fate through adhesion and regulation of signaling.

## 4. Discussion

The objective of this study was to develop surfaces containing controlled density of immobilized BMP-6 and co-present it with RGD ligands to direct cell adhesion and signaling. Cell adhesion and spreading were key elements to investigate cell differentiation and tissue repair. Several studies have demonstrated that cyclic RGD peptides promote osteogenic differentiation, especially when they participate in BMP-mediated signaling [32,33,34]. Focusing on early adhesion processes, we found that the presence of RGD ligands induced peripheral clustering of vinculin leading to the formation of FAs in C2C12 cells. Moreover, when BMP-6 was added to the media, cells displayed larger FAs and smaller dimensions, if compared to the control ones.

FAs assembly and cell spreading are both mediated by integrins, transmembrane receptors, whose cross-talk with BMPs has been investigated at several levels showing the involvement of this growth factor family in cell adhesion and migration as well as reorganization of the cytoskeleton [23,30]. In comparison to soluble BMPs the immobilization systems have the advantage of controlled and sustained influence on cell behavior, altering signaling kinetics or activating alternative signaling pathways, with consequent enhanced biological activity and stability [29,35,36]. To take advantage of the features related to growth factor presented in matrix-bound form, we immobilized BMP-6 through a heterobifunctional linker which still allows the interaction of the growth factor with its receptors. We first immobilized BMP-6 on surfaces homogeneously covered with gold and proved its stability and preservation of its biological properties looking at the activation of the Smad signaling pathway and the inhibition of myotube formation. Afterwards we immobilized the growth factor on nanopatterned surfaces decorated with gold nanoparticles spaced at 30, 50 or 100 nm to control BMP-6 amount and exact number of immobilized molecules [24]. Our data indicate that when BMP-6 is immobilized on the surfaces it is able to induce short-term signaling pathways through the phosphorylation of Smad1/5/8 and it is also still active after 4 days of culture inhibiting myotube formation. We found that BMP-6 application is significantly effective even at amounts as low as 1 ng.

The results are consistent with our previous works on BMP-2 [23,24], in which we showed the activation of short and long-term signaling pathways by the immobilized growth factor. Nevertheless, the present study goes forward, since it combines the immobilization strategy with the presentation of adhesive molecules which in turn allow cell adhesion and it also identifies in BMP-6 an application full of promises. Among the BMPs, BMP-6 usage has not yet been explored enough, but it has been demonstrated to be more efficacious in inducing osteoblast differentiation in vitro, bone formation in vivo and also to be more resistant to Noggin inhibition if compared to BMP-7 [15].

Concerning the immobilization of the BMPs, several methods have been reported for their immobilization, mainly involving BMP-2 [28] and BMP-7 full proteins and short peptide sequences, on material surfaces [36,37]. More recently, site-specific modification of recombinant human BMP-2 for the covalent binding to a linker via click chemistry was immobilized to NeutrAvidin-agarose beads [38]. While our strategy does not allow a precise control over the orientation of the molecule, it however ensures the specific binding of growth factors on the surface without further modification of active residues of the protein. Compared to homogenously coated gold substrates, the nanopatterned surfaces allowed the copresentation of adhesion ligands addressing integrins. Regarding the integrin types, which might cooperate with BMP-6 in signaling pathways, results from different studies on BMP-2 show the possible involvement of β_1_ integrins, when adhesion is promoted by ECM present in the system [29,39], whereas β_3_ integrins appear to be involved in the crosstalk when integrin-mediated adhesion relies on the endogenous production of ECM by the cells cultured on the materials [30]. Addressing integrin specificity and the crosstalk with BMPs in C2C12 is of particular interest for differentiation studies on regenerative muscle, being β_3_ integrin upregulation required for myotube formation during C2C12 differentiation [40]. Therefore, it. would be interesting in future studies to define how BMP-mediated and β_3_ integrin-mediated signaling are finely regulated for myogenic or osteogenic differentiation. In a recent study, to mimic the cell environment and determine the effect of RGD and BMP-2 mimetic peptide crosstalk, the two ligands have been presented at varying surface concentrations to human bone marrow mesenchymal stem cells to trigger adhesion and osteogenic differentiation [33]. Thus, the results presented here could be also extended in the future to further stimulate osteogenic differentiation of mesenchymal stem cells and improve the coatings of artificial materials for treatment of bone defects. Moreover, our innovative approach may provide a basis for a better understanding of several biological questions related to cell response and cell fate to co-presented factors at the nanoscale, which could be interesting for biomaterial engineering in future.

## 5. Conclusions

In this work, we developed a platform for the copresentation of growth factors, namely BMP-6, and the integrin binding RGD peptide to modulate cell adhesion and signaling responses. Both extracellular ligands are covalently immobilized to prevent internalization, and therefore, achieve their sustained presentation to cells. Moreover, using surface nanopatterning, we immobilized BMP-6 and RGD in amounts as low as 1 ng, while maintaining their biological activity. Our results suggest that a local control of signals and density of ligands is crucial for modulating cell response. To our knowledge, this is the first platform that is based on the covalent immobilization of recombinant human BMP-6 and which studies the effects of this growth factor on integrin-mediated adhesion in differentiation. In future studies, the molecular mechanisms should be addressed, in particular regarding the regulation of lineage commitment and differentiation of cells.

## Figures and Tables

**Figure 1 cells-08-01646-f001:**
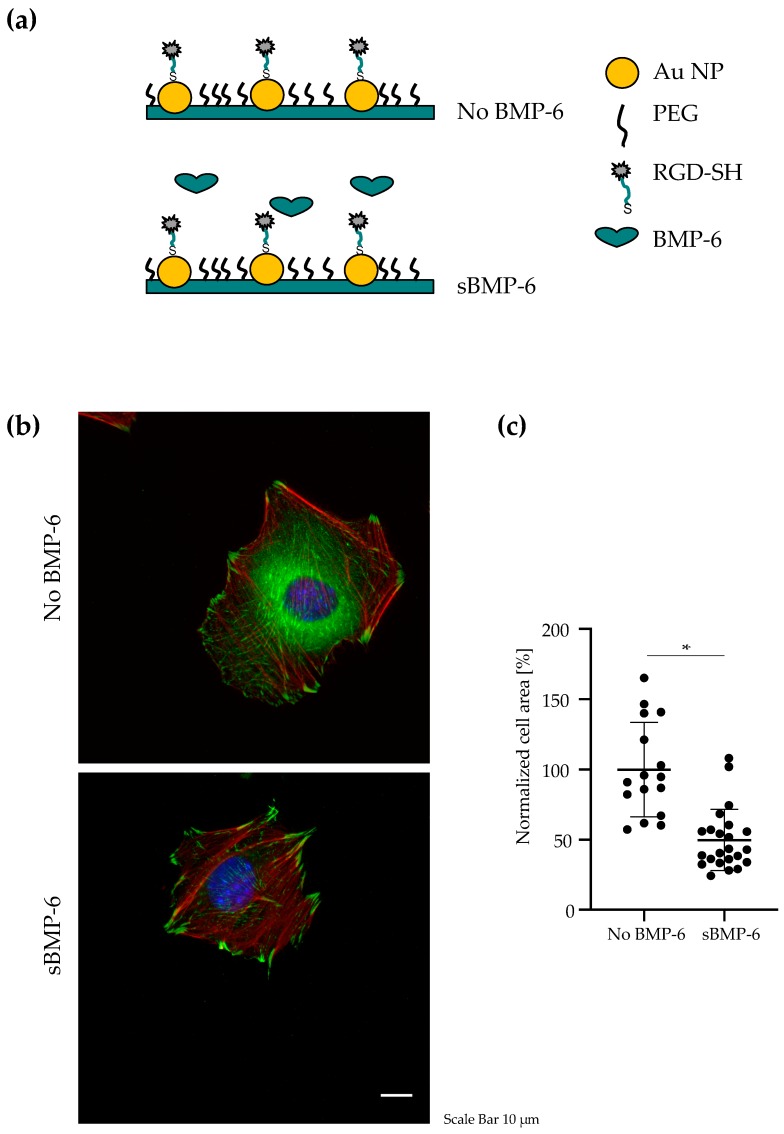
Effects of BMP-6 on adhesion of C2C12 cells. (**a**) The space between the gold nanoparticles (interparticle distance of 50 nm) is passivated using the PEG2000, then RGD ligands carrying a -SH group are immobilized on the nanoparticles to promote integrin-mediated cell adhesion. As control (No BMP-6), cells were not treated with the growth factor. For samples incubated with BMP-6 (sBMP-6), the molecule was added to the media at a concentration of 10 nM. (**b**) Representative immunofluorescence images of vinculin (green), nuclei (blue) and actin (red) in C2C12 cells. Cells were adhering for 4 h to nanostructured glass surfaces without BMP-6 or with sBMP-6. Scale bar: 10 μm. (**c**) Cell area normalized to the average cell area of the control samples plotted as means ± SD of at least 17 cells imaged for each treatment in 3 independent repeats, * *p* < 0.001. Student’s *t*-test was used for comparison.

**Figure 2 cells-08-01646-f002:**
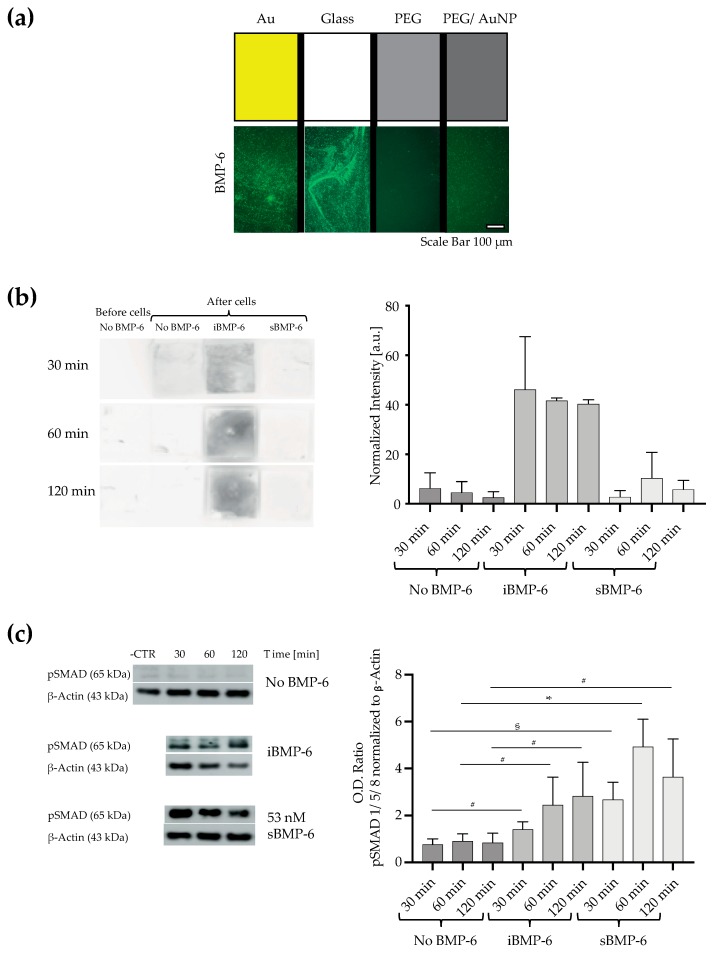
Surface immobilized BMP-6 is not removed by cells and it retains its biological activity. (**a**) Indirect immuno-detection of BMP-6 bound to surfaces. Different underlying coating of glass coverslips are incubated with the MU-NHS linker and BMP-6. The growth factor is detected by imaging the binding of anti-BMP-6 antibodies and Alexa488-conjugated antibodies (fluorescence microscopy imaging). Scale bar: 100 μm. (**b**) C2C12 cells were treated from the top with gold homogeneous surfaces in absence of BMP-6 (control, No BMP-6), in presence of covalently immobilized BMP-6 (iBMP-6) or soluble BMP-6 (sBMP-6). BMP-6 was detected by immuno-chemiluminescence before incubation of the surfaces with cells (before cells), and after 30, 60 and 120 min of incubation with cells. The graph represents means ± SD of two independent repeats. (**c**) After 30, 60 and 120 min of incubation with the surfaces, cell lysates were immunoblotted for pSMAD 1/5/8 and β-actin. Data are presented as mean ± SD of three independent experiments # *p* < 0.05, § *p* < 0.01, * *p* < 0.001. Student’s *t*-test was used for single comparison.

**Figure 3 cells-08-01646-f003:**
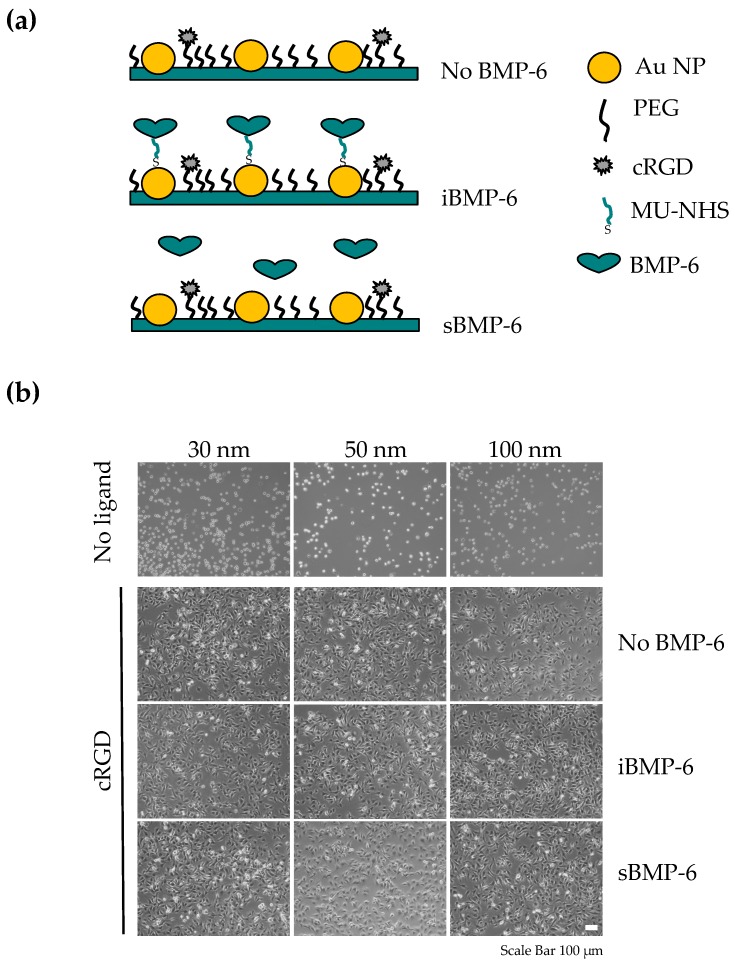
Design of the platform for the co-presentation of iBMP-6 and RGD ligands to control cell adhesion. (**a**) The PEG layer between gold nanoparticles on the surface is functionalized by click chemistry with RGD ligands (No BMP-6, used as negative control); then, BMP-6 is covalently immobilized on the gold nanoparticles (iBMP-6) or added to the media (sBMP-6). (**b**) Cell adhesion of C2C12 cells to nanopatterned surfaces co-presenting iBMP-6 and RGD ligands. Representative phase contrast images of cells 3 h after seeding on nanostructured glass surfaces used as control (30, 50 and 100 nm spacing) without adhesion ligand or functionalized with cRGD. C2C12 cells seeded without BMP-6 (No BMP-6), with BMP-6 immobilized on the surfaces (iBMP-6) or exposed to BMP-6 added to the media (sBMP-6). Scale bar: 100 μm.

**Figure 4 cells-08-01646-f004:**
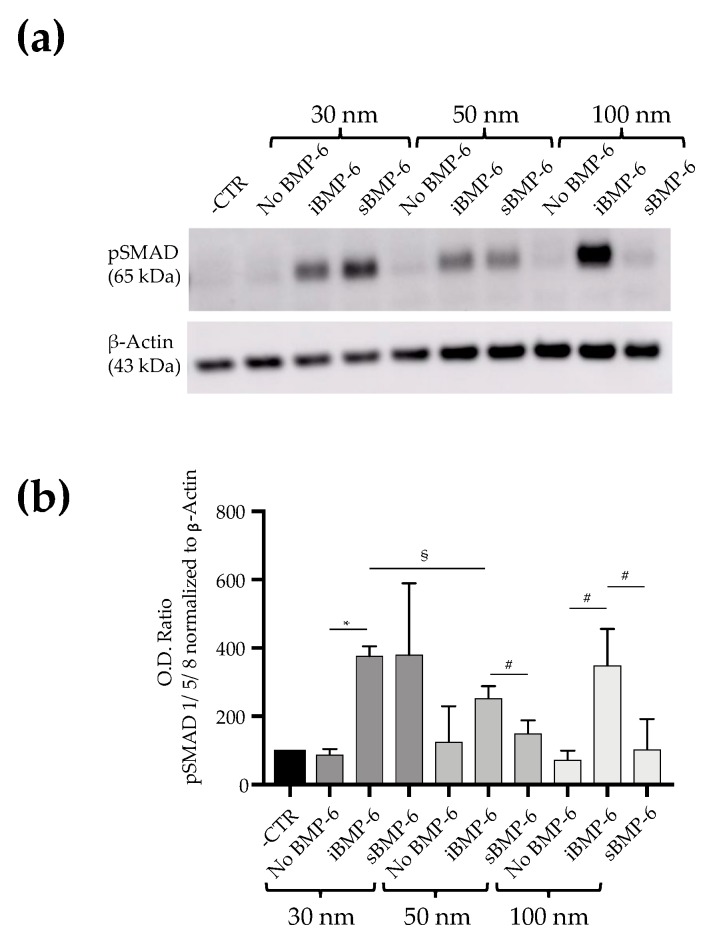
Analysis of Smad 1/5/8 phosphorylation. (**a**) Lysates from C2C12 cells 3 h after seeding were immunoblotted for pSmad1/5/8 and β-actin. (**b**) Smad1/5/8 phosphorylation levels were analyzed by Western blot and quantified by measuring the intensities of protein bands normalized to β-actin levels. Experiments were performed three times and representative images are shown, in the graphs error bars are SD, # *p* < 0.05, § *p* < 0.01, * *p* < 0.001. Student’s *t*-test was used for single comparison.

**Figure 5 cells-08-01646-f005:**
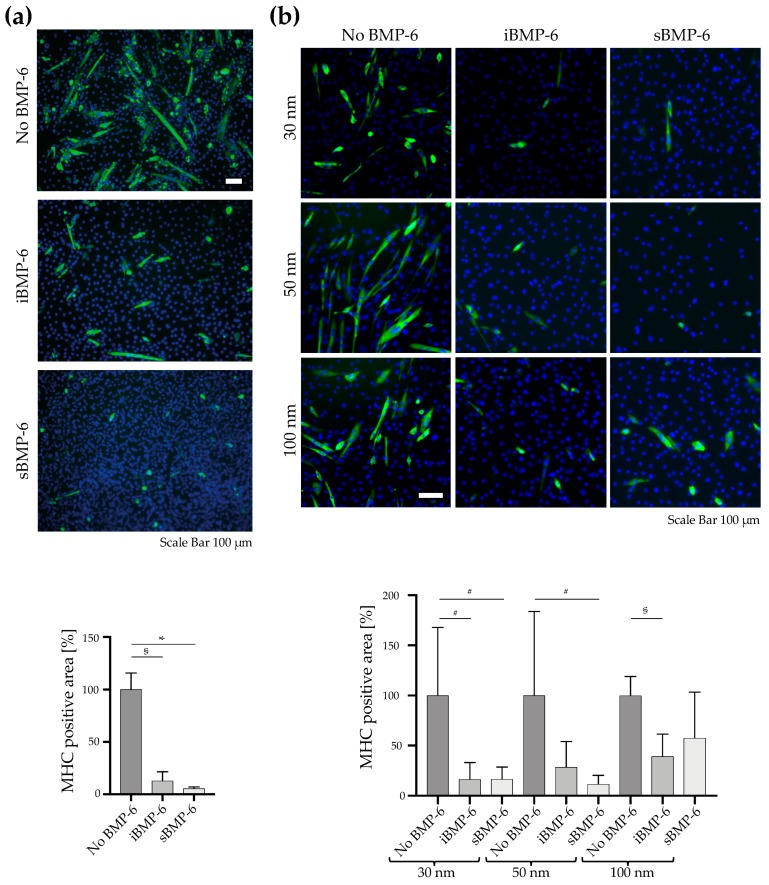
Analysis of myotubes formation in C2C12 cells. Cells were seeded on gold homogeneous surfaces (**a**) or nanopatterned surfaces (**b**) and cultured to allow myotube formation. Immunofluorescence microscopy images show myosin heavy chain (MHC) staining of multinucleated myotubes (green) and DAPI nuclei staining (blue). Myotube formation is observed when cells are cultured on control surfaces in absence of BMP-6 (No BMP-6). For iBMP-6 and sBMP-6, cells were cultured in presence of 80 ng (A) or 19, 6 and 1 ng (B) of the growth factor. MHC positive areas were quantified by image analysis. In the graphs error bars are SD, # *p* < 0.05, § *p* < 0.01, * *p* < 0.001. Student’s *t*-test was used for single comparison.

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
