# Peer review of "Copresentation of BMP-6 and RGD Ligands Enhances Cell Adhesion and BMP-Mediated Signaling"

_cells, 2019, doi:10.3390/cells8121646_

Round 1
Reviewer 1 Report
Posa et al. have established a setup to mimic matrix bound BMP6 and investigates the effect of BMP6 in this context on myoblast to osteogenic differentiation.
The difference in ligand potency when comparing soluble versus immobilized BMP6 is intriguing to me and really highlights the relevance of physiological context to the field.
I advise that the manuscript is acceptable for publication in Cells if comments below are addressed.
To investigate the specificity of the RGD is there a negative control setup that can be designed? I.e are there non-functional RGDs that can be tested in combination with BMP6?
Have the authors tested the longevity of the immobilized stimulation upon the cellular properties? For instance, if you remove the BMP6/RGD immobilized platform and grow them on RGD only, how long do they keep up pSmad levels and enhanced FA formation?
Reviewer 2 Report
Journal: Cells
Title: Copresentation of BMP-6 and RGD Ligands Affects
Adhesion and SMAD Signaling.
Authors: F. Posa, A. L. Grab, V. Martin, D. Hose, A. Seckinger, G. Mori,
S. Vukicevic and E. A. Cavalcanti-Adam
OVERALL COMMENTS/DECISION:
The paper studies the development of surfaces containing controlled density of immobilized BMP-6 and RGD ligands via a dual functionalization protocol to direct cell adhesion and signaling. Using the developed immobilization strategy the authors show that both ligands are covalently immobilized and maintain their biological activity.
The work presented in this paper is a continuation of the work of the research group on the development of surfaces with immobilized BMP-2 and the successful activation of short-term and long-term pathways by the immobilized growth factor [T. Pohl et al. Surface immobilization of bone morphogenetic protein 2 via a self-assembled monolayer formation induces cell differentiation, ActaBiomaterialia 2012; E. Schwab et al. Nanoscale Control of Surface Immobilized BMP-2: Toward a Quantitative Assessment of BMP-Mediated Signaling Events Nanoletters 2015] and as a whole exhibits a very good scientific background.
The problem statement is well explained in the Introduction section. The Materials and Methods section includes all the necessary experimental information of the study regarding the protocols and the experiments. The inclusion of an illustration of the design (Figure 1a and 3a) aids in understanding of the protocols. The Results are presented in a clear manner, with each subsection having a clear conclusion. Generally, the study is well designed with the appropriate control groups in each case (e.g. soluble BMP-6, absence of ligand, etc.). The Discussion section correlates well the findings of the study with the related literature. One comment is that the Discussion section should be enriched with the future perspectives of this experimental cell culture model/future work based on the present findings (e.g. cell type).
In the following section, there are some points that have to be corrected or rephrased.
1. Page 2, Lines 44: Please correct ‘BMP-6’ into ‘Thus,BMP-6’.
2. Reference 8: Please correct ‘BMP et cancer-’ into ‘BMP et cancer:’.
Reviewer 3 Report
The paper entitles “Copresentation of BMP-6 and RGD Ligands Affects Adhesion and SMAD Signaling” covers an interesting study of BMP-6 effect on cell adhesion. It is a clearly written paper that requires some minor revision.
In Fig 2 a) the fluorescence images should be enlarged Do you have any SEM images of cells attached to the modified substrate? 5 should be rearranged, it looks in the present form very unprofessional. The conclusions should be correlated with the previous work on BMP-6 and clearly underline the novelty in this study Consider improving the title of this paper.Author Response
Please see the attachment.
